# Structural Reliability Analysis by Using Non-Probabilistic Multi-Cluster Ellipsoidal Model

**DOI:** 10.3390/e24091209

**Published:** 2022-08-29

**Authors:** Kun Li, Hongwei Liu

**Affiliations:** 1School of Mechatronics Engineering, Changsha University, Changsha 410083, China; 2College of Mechanical and Vehicle Engineering, Hunan University, Changsha 410082, China

**Keywords:** non-probabilistic reliability analysis, multi-cluster ellipsoidal model, Gaussian cluster analysis, second order approximation method

## Abstract

Uncertainties are normally unavoidable in engineering practice, which should be taken into account in the structural design and optimization so as to reduce the relevant risks. Yet, the probabilistic models of the uncertainties are often unavailable in the problems due to the lack of samples, and the precision of the conventional non-probabilistic models are not satisfactory when the samples are of multi-cluster distribution. In view of this, an improved method by using a non-probabilistic multi-cluster ellipsoidal model (multi-CEM) for the critical structural reliability analysis is proposed in this paper, which describes the samples in a more accurate and compact way and helps to acquire more satisfactory reliability analysis results. Firstly, a Gaussian mixture model (GMM) is built for the multi-cluster samples with performing expectation maximization (EM) algorithm, based on which the multi-CEM can be constructed. In the structural reliability analysis, two cases, respectively, considering whether the components of the multi-CEM are intersected or not are researched in detail. The non-probabilistic reliability (NPR) indexes for each component of the multi-CEM are computed using the Hasofer–Lind–Rackwitz–Fiessler (HL-RF) algorithm, and then the multidimensional volume ratios of the safe domain to the whole uncertainty domain are computed based on these indexes, indicating the structural NPR. In the end, two numerical examples and a practical application are conducted and analyzed to testify the effectiveness of the method.

## 1. Introduction

In engineering practice, uncertainties are widely and unavoidably existed in structural dimensions, environmental interference, material properties, etc., which is one of the main factors that cause instability and even failure of structural performance. With the increasing demands for lightweight and efficient structures in the industry, the structural reliability analysis considering the uncertainties turns to be more and more important. To deal with the structural uncertainties, probabilistic models based on mathematical statistics have been frequently used, and a series of reliability analysis methods [1,2,3,4,5] using these kinds of models have been proposed. The probability methods use the accurate probability distribution function (PDF) to describe the uncertainty of the parameters, and evaluates the reliability and safety of the structure through the calculation of the probability statistics theory. However, the determination of the parameter PDF in the probability model needs to be based on a large amount of experimental sample data, which is extremely expensive or even impossible to obtain for many practical engineering problems. For this reason, it is usually necessary to make some subjective assumptions about the distribution characteristics of parameters in the uncertainty analysis based on the probability method. Nevertheless, some studies have shown that the small deviation of parameter probability distribution can lead to large error of uncertainty analysis results [6]. To this end, to investigate efficient structural reliability analysis methods using non-probabilistic models is of great significance.

Non-probabilistic model is acknowledged to be proposed by Ben Haim in 1990s [7]. Different from probabilistic models, non-probabilistic models usually treat the variables as uncertain and bounded, and utilize a convex set to encircle them. In the uncertainty modeling, the variation boundaries of variables can be obtained according to the engineering experience or a small number of samples. Owing to this advantage in dealing with uncertainties, non-probabilistic models have been widely researched and subsequently applied in fields of uncertainty propagation [8,9], optimization design [10,11,12], dynamic load identification [13], and especially structural reliability analysis. By formulating reliability in terms of a maximum uncertainty degree that structural parameters can tolerate, the concept of non-probabilistic reliability was then proposed [14]. Proceed in this concept, a reliability measure was further developed [15], and similar concepts were introduced into the non-probabilistic convex models according to the traditional probabilistic reliability methods. The core of this kind of model was to envelop the given sample information with the minimum bounded convex set boundary to measure its uncertainty. The traditional non-probabilistic convex model is interval model, that is, the uncertainty of variables is represented by the upper and lower bounds of samples. For example, Wang and Fang [16] proposed an interval structural reliability analysis approach based on the traditional first order reliability method, in which the NPR index was formulated based on an infinite norm for the interval convex model, and the NPR index are calculated by adopting the HL-RF algorithm [17] or the adaptive chaos control method [18]. However, the interval model cannot consider the correlation of samples, resulting in the conservative results of uncertainty measurement. In order to effectively consider the direct correlation of samples, Jiang et al. [19,20] proposed a correlation analysis technique constructing a multidimensional ellipsoidal model, and introducing the first order approximation method (FOAM) and the second order approximation method (SOAM) into NPR analysis. Liu et al. [21] suggested a pseudo-probabilistic measure method which combined the multidimensional volume ratio with the first-order approximation of the system-state function. Furthermore, by formulating the constraints in terms of NPR index, a series of reliability-based design and optimization methods have also been proposed [22,23,24]. Yet, although structural reliability analysis methods using the ellipsoidal model have been developed and enriched by many researchers, the distribution property or clustering status of uncertainty samples was rarely considered. This may lead to failure of the non-probabilistic convex models to describe the uncertain variables with multi-cluster property effectively. Meanwhile, the calculated results of reliability analysis may also be misleading, and then a risky or over conservative design would be obtained. To address this issue, it is quite necessary to construct more effective NPR models, and then develop feasible reliability analysis approaches based on these models.

In this paper, on the basis of the previous research work, the distribution property or clustering status of samples are taken into account and a multi-CEM model is built to encircle the samples in a more accurate and compact way. In addition, a quantitative model of reliability analysis is established, and the complex scenarios of ellipsoid intersection are discussed, which helps to obtain more satisfactory structural reliability analysis results. In order to realize reasonable quantification of the distribution status of structural uncertainties, Gaussian cluster analysis is introduced to build the optimum GMM for the samples. The critical elliptical contour feature of the GMM is calculated by EM algorithm and utilized to construct the multi-CEM. Before conducting reliability analysis, an approximated single-ellipsoid model (AEM) is established to describe the overlapping region between the intersected components of the multi-CEM. The structural reliability indexes are computed by combining the multi-CEM with FOAM/SOAM. Finally, similar to the conventional ellipsoidal model, a ratio of the multidimensional volume between the safe domain and the whole uncertainty domain is introduced to measure the reliability of structures. The remainder of this paper is organized as follows. In Section 2, the problem is stated and the multi-CEM is constructed. In Section 3, the non-probabilistic reliability analysis of structures based on the multi-CEM is detailed. Two numerical examples are given in Section 4, and an engineering problem is analyzed in Section 5. In Section 6, some conclusion remarks are drawn.

## 2. Problem Statement and Multi-Cluster Ellipsoidal Model Construction

With the rapid development of science and technology, the designed structures are not only required to meet the functional requirements, but also expected to have high reliability. Yet, in engineering practice, due to the complexity of actual structures, the discreteness of used materials, and the manufacturing and installation errors, the physical, geometric, and boundary characteristics of the structure inevitably suffer a certain level of uncertainty. If the relevant variables of the structure are still regarded as unique and deterministic, the designed structure may have a significant deviation from the expected one, resulting in weakening of its service effect, shortening of service life and even failure of basic functions. Therefore, to perform reliability analysis, with taking into account the uncertain variables, before structural design turns to be quite necessary and valuable.

To describe the uncertainties, probabilistic methods are acknowledged to be the most accurate ones. However, these kinds of methods require a large number of samples to compute the probability density functions, which is often unfeasible in practical engineering. In order to make up for the shortcomings of traditional probability methods, non-probabilistic methods including interval methods, polygon methods, and ellipsoid clustering methods, have been extensively studied. Yet, it is not difficult to see that they generally failed to describe the uncertainty domain in a compact and accurate way when processing variables with correlation and multi-cluster distribution characteristics. This leads to that though the designed structure can meet the basic functional requirements, it is probably not the most lightweight and efficient one. As shown in Figure 1, although the samples of the uncertain variables are of limited sizes, they still have complex distributions and multiple clustering characteristics. Apparently, to describe these uncertainty domains using interval models or single-ellipsoid models is not that appropriate.

In order to overcome the above problems, a multi-CEM is to be constructed, based on which structural reliability analysis will be performed and more accurate results can be obtained. In the cluster analysis, the PDFs of the variable with multi-cluster distributed samples can be obtained by using the GMM [25,26,27], which is approximated as the following form with multiple Gaussian functions,
(1)$$f\left(X|\partial \right)=\sum_{k=1}^{K}\alpha_{k}\Phi \left(X|\mu_{k},\Sigma_{k}\right)$$
where $\partial =\left[\alpha_{1},\cdots ,\alpha_{K},\mu_{1},\cdots ,\mu_{K},\Sigma_{1},\cdots ,\Sigma_{K}\right]$ represent the characteristic parameters of the GMM, f(X|∂) is the PDF of variable X under the parameters ∂, *K* denotes the cluster number, $\alpha_{k}$ marks the weighting coefficient of the *k*-th cluster, $0<\alpha_{k}<1$, $\sum_{k=1}^{K}\alpha_{k}=1$, and $\Phi \left(X|\mu_{k},\Sigma_{k}\right)$ refers to its Gaussian PDF with mean vector $\mu_{k}$ and covariance matrix $\Sigma_{k}$. The optimal parameters ∂ for the GMM can be computed by expectation maximization algorithm which mainly consists of E-step and M-step [28,29]. E-step calculates the probability of each point generated by different components in the mixed model and M-step adjusts the model parameters to maximize the possibility of the model generating these parameters. The determined parameters of Equation (2) are as follows,
(2)$$\left\{ \begin{array}{l} \alpha_{k}=\frac{\sum_{r=1}^{m}\beta_{k}\left(X^{\left(r\right)}\right)}{m}\\ \mu_{k}=\frac{\sum_{r=1}^{m}\beta_{k}\left(X^{\left(r\right)}\right)X^{\left(r\right)}}{\sum_{r=1}^{m}\beta_{k}\left(X^{\left(r\right)}\right)}\\ \Sigma_{k}=\frac{\sum_{r=1}^{m}\beta_{k}\left(X^{\left(r\right)}\right){\left(X^{\left(r\right)}-\mu_{k}\right)}^{T}\left(X^{\left(r\right)}-\mu_{k}\right)}{\sum_{r=1}^{m}\beta_{k}\left(X^{\left(r\right)}\right)} \end{array}\right.$$
where $X^{\left(r\right)}$ represents the *r*-th set of *n*-dimensional points, and r=1,2,⋯,m; $\beta_{k}\left(X^{\left(r\right)}\right)$ is an introduced notation which is often called as the post probability and has the form of $\beta_{k}\left(X^{\left(r\right)}\right)=\frac{\alpha_{k}\Phi \left(X^{\left(r\right)}|\mu_{k},\Sigma_{k}\right)}{\sum_{i=1}^{K}\alpha_{i}\Phi \left(X^{\left(r\right)}|\mu_{i},\Sigma_{i}\right)}$, k=1,2,⋯,K. By substituting the parameters in Equation (2) into Equation (1), the GMM can be expressed as
(3)$$f\left(X\right)=\sum_{k=1}^{K}\frac{\alpha_{i}}{{\left(2\pi \right)}^{n/2}{\left|\Sigma_{k}\right|}^{1/2}}\exp\left(-\frac{1}{2}{\left(X-\mu_{k}\right)}^{T}\Sigma_{k}{}^{-1}\left(X-\mu_{k}\right)\right)\;$$
Observing Equation (3), it can be found that the exponential term of the GMM indeed has the same form with the characteristic matrix of ellipsoid model [19]. Therefore, it is introduced to establish the corresponding ellipsoid models for the uncertain variables. It is worth mentioning here that although the accuracy of the PDF obtained by the GMM cannot be guaranteed, especially when there exists a shortage of samples, $\mu_{k}$ and $\Sigma_{k}$ which reflects the center and correlation of the ellipsoid are the two required aspects for ellipsoid modeling of each cluster. In this way, the whole uncertainty domain can be expressed as the composition form of the *K* ellipsoidal models as follows
(4)$$\Gamma_{X}=\left\{X|\overset{K}{\underset{k=1}{\cup }}{\left(X-\mu_{k}\right)}^{T}\Sigma_{k}{}^{-1}\left(X-\mu_{k}\right)\leq R_{k}^{2}\right\}$$
where $R_{k}$ denotes the *k*-th critical contour ellipsoid, which can be determined according to the critical elliptical contour feature of the GMM to ensure that the samples of each cluster are entirely encircled by the ellipsoid. Finally, the multi-CEM of the uncertainties can be constructed as the following Equation (5).
(5)$$\Gamma_{X}=\left\{X|\overset{K}{\underset{k=1}{\cup }}{\left(X-\mu_{k}\right)}^{T}\Omega_{k}\left(X-\mu_{k}\right)\leq 1\;\right\}$$
where $\Omega_{k}=\Sigma_{k}{}^{-1}/R_{k}^{2}$.

Based on the constructed multi-CEM, the distribution and clustering feature of the uncertain samples can be revealed. Meanwhile, the correlation of the uncertain variables can also be properly calculated as by using a traditional ellipsoidal model [19,20]. As shown in Figure 1, the traditional approach utilizes a single-ellipsoidal model (in red color) to envelop these samples without considering the distribution of samples. Compared to the multi-CEM, it makes the uncertain domain of variables enlarged, which may lead to the analysis results conservative and imprecise. Obviously, the multi-CEM can describe the samples of uncertain variables in a better way.

## 3. Structural Reliability Analysis Scheme Based on the Multi-CEM

According to the sample distribution of the uncertain variables, two typical situations can be considered for the constructed multi-CEM. One is that there is no intersection between the components of the multi-CEM, as shown in Figure 1a, and the other is that there is an intersection between the components, as shown in Figure 1b. In this section, the two situations will be investigated for structural reliability analysis, respectively.

### 3.1. The Situation of Non-Intersecting

As shown in Figure 2, the limit state surface divides the whole uncertainty domain into the failure domain and the safe domain. For the multi-CEM composed of multiple ellipsoidal models, there will be three cases, that is, (i) all the ellipsoidal model components fall into the failure domain, (ii) the ellipsoid model components are divided by the limit state surface, and (iii) all the components are located in the safe domain. Similar to the structural reliability analysis with single ellipsoid model, when all the components of the multi-CEM fall into the failure domain, it indicates that the structure is absolutely unreliable. On the contrary, a completely reliable structure can be obtained when all the components are located in the safe domain. If any component of the multi-CEM is divided by the limit state surface, there will be reliability risk for the structure, and the multidimensional volume ratio of the safe domain to the whole uncertainty domain can indicate the reliability degree of the structure. When the multidimensional volume ratio increases, the reliability degree will become greater.

In the reliability analysis, a key and prior step is the computation of the NPR index β for each component of the multi-CEM. In order to calculate β for each component, the limit state function should be approximated at first. When the limit state function is linear or with weak nonlinearity, accurate results can be obtained by combining the multi-CEM with the FOAM. While when the limit state function is strongly nonlinear, the SOAM [20] is suggested, so as to improve the calculation accuracy. Then by performing regularization on the multidimensional ellipsoid and eigenvalue-decomposition on the characteristic matrix $\Omega_{k}$, a unit sphere space δ can be obtained for each ellipsoidal component [19]. In each δ space, an optimization problem as below can be established.
(6)$$\left\{ \begin{array}{l} \beta =\min\left\|\delta \right\|\\ s.t.\;G\left(\delta \right)=0 \end{array}\right.$$
where ‖⋅‖ means the norm of a vector, G(δ) represents the limit state function. As shown in Figure 3, and $\delta^{*}={\left[\delta_{1}^{*},\delta_{2}^{*},\ldots ,\delta_{n}^{*}\right]}^{T}$ are the most probable points (MPP) of failure, which are located on the limit state surface and closest to the original point in δ space.

To resolve the optimization problem of Equation (6), the HL-RF iterative algorithm [30,31] is adopted, and the results are obtained as Equation (7).
(7)$$\beta^{\left(i\right)}=\frac{G\left(\delta^{\left(i\right)}\right)-\nabla G{\left(\delta^{\left(i\right)}\right)}^{T}\delta^{\left(i\right)}}{\left\|\nabla G\left(\delta^{\left(i\right)}\right)\right\|}$$
where
(8)$$\left\{ \begin{array}{l} \delta^{\left(i+1\right)}=\delta^{\left(i\right)}+\lambda d^{\left(i\right)}\\ d^{\left(i\right)}=\frac{\nabla G{\left(\delta^{\left(i\right)}\right)}^{T}\delta^{\left(i\right)}-G\left(\delta^{\left(i\right)}\right)}{{\left\|\nabla G\left(\delta^{\left(i\right)}\right)\right\|}^{2}}\nabla G\left(\delta^{\left(i\right)}\right)-\delta^{\left(i\right)} \end{array}\right.$$
The superscript (*i*) represents the *i*-th iteration step, and ∇G(δ) marks the gradient vector of G with respect to δ.

With the calculated index β, the reliability of this component can then be obtained as below. For expression simplicity, the 2-dimensional uncertain variables are employed. When the principal curvature $k^{*}\neq 0$, the limit state function around ${\left[\delta_{1}^{*},\delta_{2}^{*},\ldots ,\delta_{n}^{*}\right]}^{T}$ can be approximately expressed by its second-order Taylor expansion as the following Equation (9), which is shown in Figure 4.
(9)$$G\left(\delta \right)\approx G\left(\delta^{*}\right)+{\left(\delta -\delta^{*}\right)}^{T}\nabla G\left(\delta^{*}\right)+\frac{1}{2}{\left(\delta -\delta^{*}\right)}^{T}\nabla^{2}G\left(\delta^{*}\right)\left(\delta -\delta^{*}\right)$$
where $\nabla G\left(\delta^{*}\right)$ and $\nabla^{2}G\left(\delta^{*}\right)$, respectively, denote the gradient vector and Hessian matrix of G with respect to $\delta^{*}$.

According to Ref. [20], the position of the characteristic plane in Figure 4 can be calculated as
(10)$$Y_{n}^{*}=\frac{-1+\sqrt{{\left(k^{*}+\beta \right)}^{2}+1-\beta^{2}}}{k^{*}}$$
For the cases shown in Figure 4, the volumes of failure domain with convex surface and concave surface are, respectively,
(11a)$$A_{1}=A_{\mathrm{cap}}+A_{\mathrm{pan}}$$
(11b)$$A_{1}=A_{\mathrm{cap}}-A_{\mathrm{pan}}$$
Further, the reliability of the ellipsoidal convex model can be calculated as [20]
(12)$$re=\left\{ \begin{array}{ll} 0, & \beta <-1,\\ \frac{1+sign(Y_{n}^{*})}{2}-sign(Y_{n}^{*})\frac{1}{2}I_{1-{\left(Y_{n}^{*}\right)}^{2}}\left(\frac{n+1}{2},\frac{1}{2}\right)-\frac{k^{*}}{n+1}\frac{1}{B\left(\frac{n+1}{2},\frac{1}{2}\right)}{\left[1-{\left(Y_{n}^{*}\right)}^{2}\right]}^{\frac{n+1}{2}}, & -1\leq \beta \leq 1,\\ 1, & \beta >1. \end{array}\right.$$
where sign(⋅) and $B_{\cdot }\left(\cdot ,\cdot \right)$ denote the sign function and Beta function, respectively, $I_{\cdot }\left(\cdot ,\cdot \right)$ refers to the regularized incomplete Beta function. When the principal curvature $k^{*}=0$, the intersecting surface of the unit hypersphere and the limit state surface is infinitely close to the (*n* − 1)-dimensional hyperplane $Y_{n}^{*}=\beta$. Thus, the reliability of each ellipsoidal model can be calculated as
(13)$$re=\left\{ \begin{array}{l} 0,\\ \frac{1+sign\left(\beta \right)}{2}-sign\left(\beta \right)\frac{1}{2}I_{1-\beta^{2}}\left(\frac{n+1}{2},\frac{1}{2}\right),\\ 1, \end{array}\right.\begin{array}{l} \beta \leq -1,\\ -1<\beta <1,\\ \beta \geq 1. \end{array}$$

Because the regularization of the multidimensional ellipsoids and the eigenvalue decomposition of matrices $\Omega_{k}$ are both linear transformations, so, for the components of the multi-CEM, the volume ratios of the safe domain to the whole uncertainty domain in the δ space are similar to that in the X space. Thus, by denoting the volume ratio of the *h*-th component as $re_{h}^{d}$, the non-probabilistic reliability of structure based on the multi-CEM in the X space can be calculated as
(14)$$RE=\frac{V_{safe}}{V_{total}}=\frac{\sum_{j=1}^{N_{1}}V_{j}^{s}+\sum_{h=1}^{N_{3}}\left(re_{h}^{d}\times V_{h}^{d}\right)}{\sum_{i=1}^{N_{1}}V_{i}^{s}+\sum_{j=1}^{N_{2}}V_{j}^{f}+\sum_{h=1}^{N_{3}}V_{h}^{d}}$$
where $V^{s}$ and $V^{f}$, respectively, denote the volumes of ellipsoidal models in the safe domain and the failure domain, and $V^{d}$ represents the volume of ellipsoidal model which is intersected by the limit state surface. $N_{1}$, $N_{2}$, and $N_{3}$ are the numbers of each kind of ellipsoidal convex models as shown in Figure 2, respectively. The volume of the multi-dimensional hyper-ellipsoid can be computed by
(15)$$V_{i}=\pi^{\frac{n}{2}}/\Gamma \left(\frac{n+2}{2}\right)\prod_{j=1}^{n}s_{j}$$
where *n* denotes the dimension of the uncertain variables, and $s_{j}$ marks the semi-axis of the hyperellipsoid in X space.

### 3.2. The Situation of Intersecting

For this situation, before conducting structural reliability analysis, a necessary step is to deal with the overlapping region of the intersected components of the multi-CEM. In this paper, the overlapping region is described by an approximated single-ellipsoid model. Taking any one of the ellipsoid model components in the multi-CEM for example, the construction of the AEM can be realized as follows. By transforming the original variable space X into the unit sphere space δ, the spherical coordinates $r,\;\theta_{1},\;\theta_{2},\;\cdots ,\;\theta_{n-1}$ can be built, where $r\in I_{1}=\left[0,\;1\right]$ and $\theta_{i}\in I_{2}={\left[0,\;2\pi \right]}^{n-1}$, i=1,2,⋯,n−1. Through making the spherical coordinates $\theta_{i}$ uniformly distribute in the (*n* − 1)-dimensional interval box $I_{2}$, and the r coordinates follow the PDF of $F\left(r\right)=r^{n}$ in the interval $I_{1}$, samples can be guaranteed to be uniformly scattered and distributed in the δ space. Plus, these samples with uniform distribution can be expressed as the following form of Equation (16) [32]. As the unit sphere space δ is obtained from the original variable space X by linear transformation, these samples are also uniformly distributed in space X.
(16)$$\delta =\left[\begin{array}{l} r\cos\theta_{1}\\ r\sin\theta_{1}\cos\theta_{2}\\ r\sin\theta_{1}\sin\theta_{2}\cos\theta_{3}\\ \vdots \\ r\sin\theta_{1}\sin\theta_{2}\ldots \sin\theta_{n-2}\cos\theta_{n-1}\\ r\sin\theta_{1}\sin\theta_{2}\ldots \sin\theta_{n-2}\sin\theta_{n-1}\cos\theta_{n} \end{array}\right]$$
Afterwards, the samples that meet the constraint of Equation (17) are extracted to construct the AEM for the overlapping region. As shown in Figure 5a, based on these scattered samples, its GMM can be obtained as Equation (18) by combining Gaussian cluster analysis with EM algorithm. Determining the optimal *R* to ensure that these samples are enveloped tightly, the AEM as shown in Figure 5b can be obtained, which can also be expressed as Equation (19) with $\Omega =\Sigma^{-1}/R^{2}$.
(17)$$X|\overset{2}{\underset{i=1}{⋂}}{\left(X-u_{i}\right)}^{T}\;\Omega_{i}\left(X-u_{i}\right)\leq 1$$
(18)$$f\left(X\right)=\frac{1}{{\left(2\pi \right)}^{n/2}{\left|\Sigma \right|}^{1/2}}\exp\left(-\frac{1}{2}{\left(X-\mu \right)}^{T}\Sigma^{-1}\left(X-\mu \right)\right)$$
(19)$$E_{X}=\left\{X|{\left(X-\mu \right)}^{T}\;\Omega \left(X-\mu \right)\leq 1\;\right\}$$

It is worth noting here that when there are multiple uncertain variables, the intersection of the multi-CEM components indeed can be complex, which, unavoidably, cannot be observed intuitively and accurately. In view of this, the components with failure risk ($-1\leq \beta_{i}\leq 1$) will be selected and combined in pairs. Through spatial transformation and scattering points, the scattered samples belonging to the overlapping region can be determined by using Equation (17). Then an approximate single ellipsoid model is established based on these sample points, and the reliability of the structure is calculated by Equation (21). Obviously, if two ellipsoids with failure risk do not intersect, the sample set determined by Equation (17) is empty, and no further consideration is required.

For the situation that the limit state surface divides the whole uncertainty domain into two parts, two cases can be, respectively, researched as shown in Figure 6. The first case is that the whole intersection is located in the safe domain as shown in Figure 6a, while the second is that the intersection is divided by the limit state surface as shown in Figure 6b.

Combining the FOAM/SOAM with the multi-CEM, and performing HL-RF iteration algorithm, the reliability index β of each component in the multi-CEM can be calculated. Then according to the calculated reliability index, the ellipsoidal models with failure domain can be figured out. For the Case I, as shown in Figure 6a, assuming that $N_{3}$ ellipsoidal models are risking failure, the reliability indexes of these models satisfy $-1\leq \beta_{h}^{d}\leq 1$, $h=1,2,\cdots ,N_{3}$. Considering that the volumes of all the AEMs are double-counted, the NPR of the multi-CEM in the X space can be calculated as follows by subtracting the volume of the recalculated AEMs from the total volume.
(20)$$RE=\frac{\sum_{i=1}^{N_{1}}V_{i}^{s}+\sum_{h=1}^{N_{3}}re_{h}^{d}\times V_{h}^{d}-\sum_{t=1}^{N_{4}}V_{t}^{a}}{\sum_{i=1}^{N_{1}}V_{i}^{s}+\sum_{j=1}^{N_{2}}V_{j}^{f}+\sum_{h=1}^{N_{3}}V_{h}^{d}-\sum_{t=1}^{N_{4}}V_{t}^{a}}$$
where $V^{a}$ and $N_{4}$, respectively, represent the volume and the number of the AEMs. For the Case II, as shown in Figure 6b, the overlapping region is divided by the limit state surface, and a part of the region falls into the failure domain. Supposing that $N_{3}$ components and $N_{4}$ AEMs have failure domain, it has $-1\leq \beta_{h}^{d}\leq 1$, $h=1,2,\cdots ,N_{3}$ and $-1\leq \beta_{t}^{a}\leq 1$, $t=1,2,\cdots ,N_{4}$. Easy to see, the volume of the safe domain of the AEMs are double counted, hence the NPR of the multi-CEM can be calculated as
(21)$$RE=\frac{\sum_{i=1}^{N_{1}}V_{i}^{f}+\sum_{h=1}^{N_{3}}re_{h}^{d}\times V_{h}^{d}-\sum_{t=1}^{N_{4}}re_{t}^{a}\times V_{t}^{a}}{\sum_{i=1}^{N_{1}}V_{i}^{s}+\sum_{j=1}^{N_{2}}V_{j}^{f}+\sum_{h=1}^{N_{3}}V_{h}^{d}-\sum_{t=1}^{N_{4}}V_{t}^{a}}$$
where $re_{t}^{a}$ denotes the reliability ratio of the *t*-th AEM.

## 4. Numerical Examples and Discussion

In order to show the details of the proposed method and prove its effectiveness, two numerical examples are conducted and discussed. In the two examples, the non-intersecting multi-CEM and the intersecting multi-CEM are, respectively, constructed. The results of the structural non-probabilistic reliability analysis are compared with that by using correlation approximate method (CAM) [20]. Moreover, Monte Carlo simulation (MCS) is performed, which provides the reference solutions.

### 4.1. Numerical Example 1—The Situation of Non-Intersection

In this example, the following limit state function is considered,
(22)$$G_{X}\left(x_{1},x_{2}\right)=2x_{1}-3x_{2}+a$$
where $x_{1}$ and $x_{2}$ are the two uncertain-but-bounded variables, and a means the threshold value. The available samples of the uncertain variables are shown in Figure 7 and listed in Table 1. It can be intuitively observed that the samples are distributed in two separate clusters.

By performing Gaussian clustering analysis, the GMM of these samples is achieved. The parameters of the GMM are computed by using the EM algorithm and are given in Table 2. In order to entirely encircle these samples, the critical elliptical contour feature $R_{1}^{2}$ and $R_{2}^{2}$ are, respectively, determined as 3.884 and 4.448. The multi-CEM is then constructed as shown in Figure 7 by the red and blue ellipses. For comparison, the correlation analysis is also conducted for these samples and the CAM is established as Equation (23), which is shown in Figure 7 as well by the yellow ellipse. Obviously, to quantify this uncertainty, the CAM uses a larger ellipse to envelop these samples and appears to be very rough, which may lead to conservative results in NPR analysis. On the contrary, the multi-CEM describes the whole uncertain domain using two small non-intersecting ellipses, which is much more reasonable.
(23)$${\left[\begin{array}{l} x_{1}-2.97\\ x_{2}-3.97 \end{array}\right]}^{T}\left[\begin{array}{l} 0.3568\;\;\;\;\;-0.2044\\ -0.2044\;\;\;0.3322 \end{array}\right]\left[\begin{array}{l} x_{1}-2.97\\ x_{2}-3.97 \end{array}\right]\leq 1$$

In this example, the limit state function of system is linear, therefore the FOAM and the HL-RF iteration are adopted to compute the NPR indexes. When the value of the limit state function is bigger than zero, it means that the design is reliable. Based on this, the reliability analysis results formulated by the multidimensional volume ratios of the safe domain to the whole uncertainty domain are computed under different thresholds, and are shown in Figure 8. It can be seen that as the threshold increases, the reliability of the structure becomes higher and higher. When the threshold a is set around 9, the structure generally turns to be completely reliable. For comparison, the structural reliability is also computed by using the CAM and MCS. In MCS, the two variables obey the uniform distribution, the median values are 3.16 and 4.02, and 10^5^ samples are used. As shown in Figure 8, the results obtained by using the multi-CEM are much closer to the results by MCS than that by using CAM, which demonstrates that the multi-CEM based structural reliability analysis is more effective for processing the uncertain variables of multi-clustering characteristics.

### 4.2. Numerical Example 2—The Situation of Intersection

In this example, the three-dimensional bounded uncertain variables $x_{1}$, $x_{2}$, and $x_{3}$ are considered, which are all assumed to follow the normal distribution in theory with the median values 3.358, 3.064, and 3.800, respectively. However, the samples of the uncertain variables are limited, which are listed in Table 3. The limit state function is of the following nonlinear form.
(24)$$G_{X}\left(x_{1},x_{2},x_{3}\right)=x_{1}^{2}x_{2}+x_{2}^{2}-x_{3}+a$$

The GMM is obtained by implementing Gaussian clustering analysis for these samples. The optimal parameters of the GMM calculated by using EM algorithm are listed in Table 4. By normalizing the scale, the intersecting multi-CEM with two components are constructed, and the multi-CEMs on the three two-dimensional planes are, respectively, shown in Figure 9. It can be seen that on the *X*_1_-*X*_2_ and *X*_1_-*X*_3_ planes, the two components of the multi-CEM are partly intersected, while on the *X*_2_-*X*_3_ plane, the component 2 is entirely contained by the component 1, which implies that the samples are of complex distribution. For comparison convenience, the CAM is also constructed within these samples by using the correlation ellipsoid modeling technology, which is shown in Figure 10. The uncertainty domain of the CAM is of the form of Equation (25). Based on the covariance matrices of the ellipsoid models, the minimum volumes of the multi-CEM and the CAM can be calculated, which are 1.8326 and 3.4702, respectively. Obviously, the CAM uses a single ellipsoid with a larger volume than multi-CEM to encircle these samples, while the multi-CEM can provide a more compact model for the uncertain variables with multi-cluster distribution.

To compute the structural reliability, considering that the two clusters are intersected, a single AEM is constructed in advance for the overlapping region. In this example, 375 samples are generated and uniformly scattered in the overlapping region by using the method illustrated in Section 3.2. The number of scattered points can be determined through an iterative process, that is, a certain number of points are scattered, an ellipsoidal model is established, and then the number of points is increased. When the ellipsoidal model tends to be stable, the number of scattered points can be determined. Through Gaussian clustering analysis and EM algorithm, the optimal Gaussian function for these additional samples is constructed, whose parameters are listed in Table 5. Via determining the critical contour ellipsoid, the AEM is then constructed.
(25)$${\left[\begin{array}{l} x_{1}-3.16\\ x_{2}-3.23\\ x_{3}-4.13 \end{array}\right]}^{T}\left[\begin{array}{c} \;\;0.3060\;\;\;\;\;\;0.2386\;\;\;\;-0.0060\\ \;\;0.2386\;\;\;\;\;\;1.0651\;\;\;\;-0.4476\\ -0.0060\;-0.4476\;\;\;\;\;\;\;0.5320 \end{array}\right]\left[\begin{array}{l} x_{1}-3.16\\ x_{2}-3.23\\ x_{3}-4.13 \end{array}\right]\leq 1$$

Viewing that the considered limit state function is nonlinear, the SOAM is adopted to approximate it. The structural reliability indexes are calculated by using HL-RF algorithm. Via adjusting the threshold, the reliability analysis results under different limit state functions are obtained. The reference solution is calculated by MCS using 10^5^ samples. As shown in Figure 11, the results obtained by using the multi-CEM are very close to the results by MCS, which is more accurate than that by using the CAM. This phenomenon once again proves the rationality and advancement the proposed method.

## 5. Application: Reliability Analysis of an Augmented Reality Glasses

The augmented reality glasses (AR glasses) are developing rapidly in recent years [33,34], and widely applied in the fields of medical treatment, education, industry, cultural tourism, security, etc., owing to its versatility in computing, communication, photography and positioning. As shown in Figure 12, the AR glasses is mainly composed of the controller, the spectacle frame, the battery, and the micro camera and projector. In the design of AR glasses, many requirements such as comfort and safety should be satisfied, however, many uncertain factors are also existed. For example, due to the uncertainty of working conditions such as the ambient temperature and the power consumption, the actual temperature response of the controller is highly uncertain. In order to ensure the operation safety and wearing comfort, reliability analysis appears to be quite necessary by taking into account the four main relevant uncertain variables, i.e., the power consumption *P*_1_ of chip A and *P*_2_ of chip B, the air velocity *V* and the environmental temperature *T*.

According to the measured samples, the uncertainty domain of the mentioned four variables is modeled as the multi-CEM with two clusters as follows.
(26a)$${\left[\begin{array}{l} T-8.25\\ V-0.19\\ P_{1}-0.19\\ P_{2}-0.06 \end{array}\right]}^{T}\left[\begin{array}{c} \;\;0.0213\;\;-1.7311\;\;-0.4767\;\;\;\;\;\;0.0513\\ -1.7311\;\;\;\;\;\;171.38\;\;\;\;\;\;34.187\;\;\;\;-27.241\\ -0.4767\;\;\;\;\;\;34.187\;\;\;\;\;\;46.025\;\;\;\;-47.313\\ \;\;0.0513\;\;-27.241\;\;-47.313\;\;\;\;\;\;365.62 \end{array}\right]\left[\begin{array}{l} T-8.25\\ V-0.19\\ P_{1}-0.19\\ P_{2}-0.06 \end{array}\right]\leq 1$$
(26b)$${\left[\begin{array}{l} T-32.50\\ V-0.455\\ P_{1}-0.410\\ P_{2}-0.133 \end{array}\right]}^{T}\left[\begin{array}{c} \;0.0373\;\;-2.4812\;\;-0.0253\;\;-0.9501\\ -2.4812\;\;\;\;\;\;183.78\;\;\;\;-5.5348\;\;59.546\\ -0.0253\;\;-5.5348\;\;\;\;34.387\;\;-29.021\\ -0.9501\;\;\;\;\;\;59.546\;\;-29.021\;\;\;\;275.07 \end{array}\right]\left[\begin{array}{l} T-32.50\\ V-0.455\\ P_{1}-0.410\\ P_{2}-0.133 \end{array}\right]\leq 1$$
In order to ensure the wearing comfort and operation safety, the temperatures of the controller at regions A and B should be controlled below a certain threshold. For this purpose, the following two limit-state functions can be established,
(27a)$$G_{A}=T_{A}^{0}-T_{A}\left(d_{1},d_{2},d_{3},T,V,P_{1},P_{2}\right)$$
(27b)$$G_{B}=T_{B}^{0}-T_{B}\left(d_{1},d_{2},d_{3},T,V,P_{1},P_{2}\right)$$
where $T_{A}^{0}$ and $T_{B}^{0}$ denote the thresholds, $T_{A}$ and $T_{B}$ are the corresponding actual temperature responses, $d_{1}$, $d_{2}$, and $d_{3}$ represent the deterministic structural sizes of the controller housing.

The simulation model of the controller is created and shown in Figure 13, in which 4 components and 22,928 thermal couplings eight-node hexahedrons are included. By calling the simulation model once, the function values of the two constraints $T_{A}$ and $T_{B}$ can be obtained once. In order to achieve parameterization and improve efficiency, the simulation model was called 100 times. According to the simulated 100 values, the second-order polynomial response surfaces of $T_{A}$ and $T_{B}$ are constructed as the following Equations (28a) and (28b). The accuracy of the constructed response surfaces is verified by comparing several randomly selected points on the response surfaces with the simulated values.
(28a)$$\begin{array}{ll} T_{A}= & 100.47-124.32d_{1}-14.39d_{2}+10.74d_{3}\\ & +1.50d_{1}d_{2}+10.10d_{1}d_{3}+0.29d_{2}d_{3}+38.32d_{1}^{2}+4.28d_{2}^{2}-8.9d_{3}^{2}\\ & +13.65P_{1}-42.59P_{2}+17.51P_{1}P_{2}+4.85P_{1}^{2}+145.24P_{2}^{2}\\ & +0.90T-34.41V-0.18TV+28.41V^{2} \end{array}$$
(28b)$$\begin{array}{ll} T_{B}= & 73.27-88.5d_{1}-11.92d_{2}+10.13d_{3}\\ & +0.54d_{1}d_{2}+6.67d_{1}d_{3}+1.15d_{2}d_{3}\;+27.81d_{1}^{2}+3.45d_{2}^{2}-7.51d_{3}^{2}\\ & +6.60P_{1}-28.69P_{2}+26.31P_{1}P_{2}+3.63P_{1}^{2}+81.92P_{2}^{2}\\ & +0.92T-34.73V-0.09TV+32.49V^{2} \end{array}$$

Combing the multi-CEM with the SOAM, and performing the HL-RF iteration algorithm, the NPR analysis is conducted under a series of thresholds to assess the reliability of the AR glasses. As shown in Figure 14, as the thresholds of $T_{A}$ and $T_{B}$ increase from −10 °C to 50 °C, the corresponding thermal performance becomes more and more reliable. Though the four parameters are uncertain, when the thresholds of $T_{A}^{0}$ and $T_{B}^{0}$, respectively, reach 39 °C and 42 °C, the thermal performance becomes completely reliable. Meanwhile, the results obtained through the proposed method is very close to that by using MCS, indicating that the proposed method based on the multi-CEM is suitable for analyzing the NPR of AR glasses.

## 6. Conclusions

An effective non-probabilistic reliability analysis method based on the multi-cluster ellipsoidal model is presented. A multi-CEM is constructed for the samples according to the ellipsoid critical contour feature of the GMM, which can deal with the multi-cluster distribution characteristics of the uncertain variables. The FOAM/SOAM is utilized to approximate the limit state functions, with which the NPR index of each multi-CEM component can be computed through HL-RF algorithm. The reliability of the studied structure is quantified by the multi-dimensional volume ratio of the safe domain to the whole convex domain. Two numerical examples and an engineering application are conducted in the end validating the effectiveness of the proposed method. Compared to the traditional convex model, the constructed multi-CEM has a rigorous but understandable form, and is more effective for handling the uncertainty with complex distribution.

## Figures and Tables

**Figure 1 entropy-24-01209-f001:**
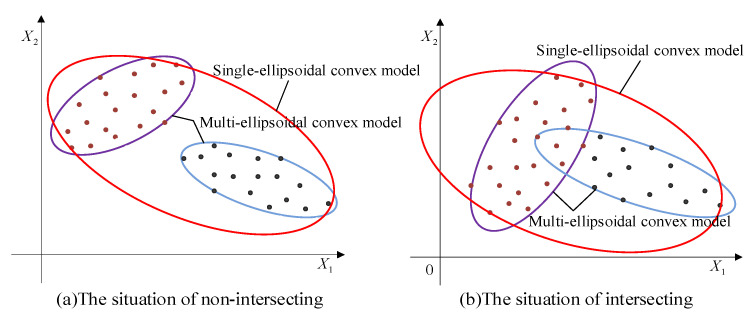
The description of the two-dimensional uncertain variables.

**Figure 2 entropy-24-01209-f002:**
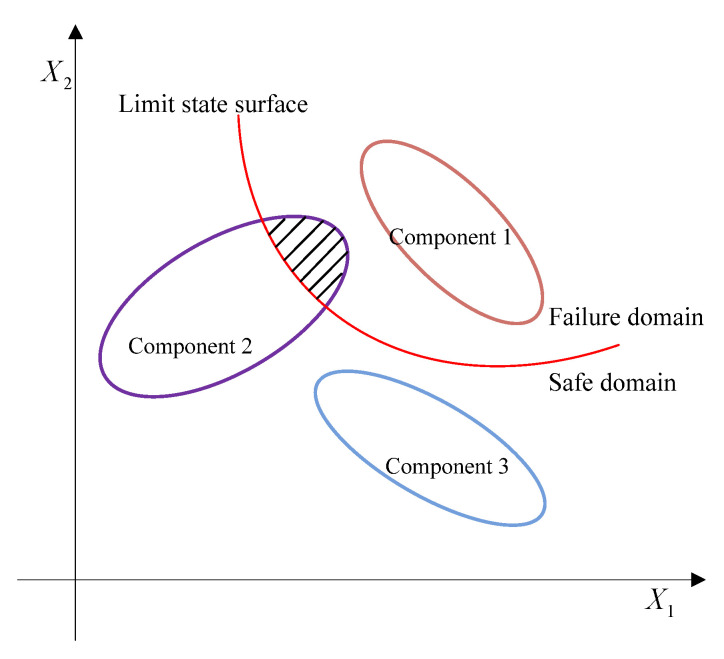
The distribution of components of the multi-CEM.

**Figure 3 entropy-24-01209-f003:**
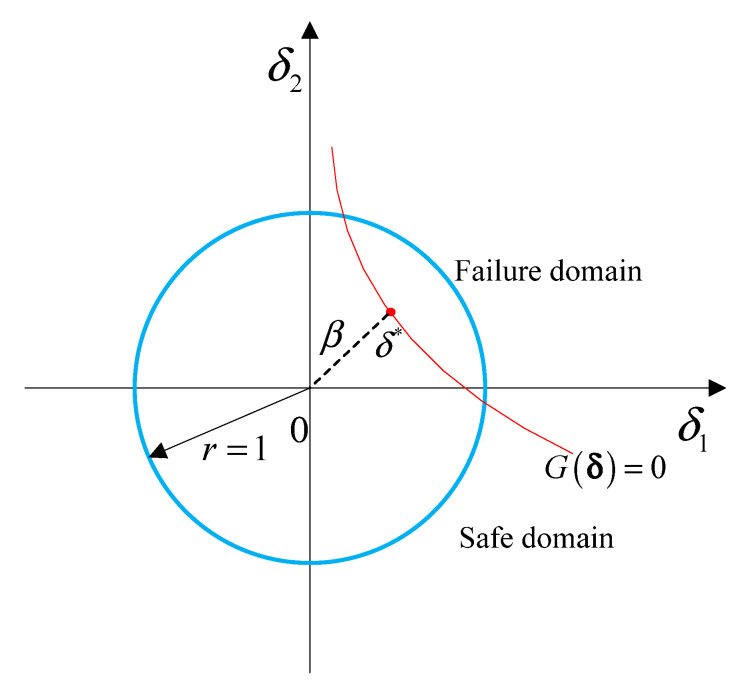
Non-probabilistic reliability index and MPP in the unit sphere space.

**Figure 4 entropy-24-01209-f004:**
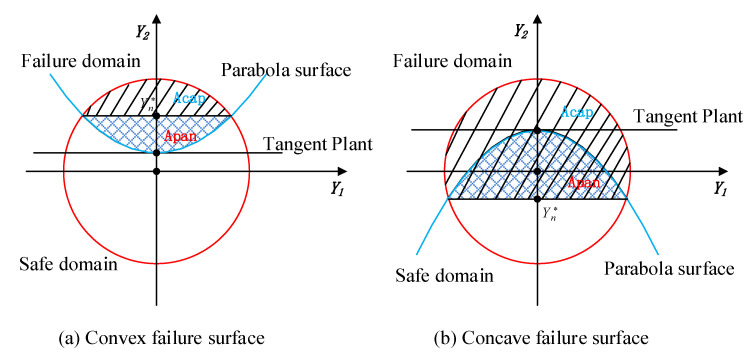
Division of the uncertainty domain by convex and concave surfaces.

**Figure 5 entropy-24-01209-f005:**
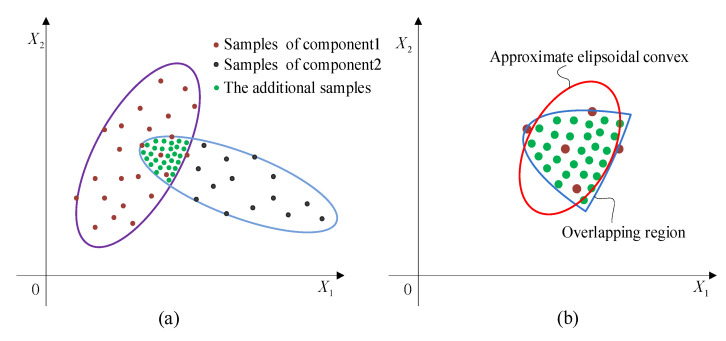
Sampling and construction of the approximate ellipsoid model ((**a**) The uniformly scattered samples in the overlapping region; (**b**) the established approximated single-ellipsoid model).

**Figure 6 entropy-24-01209-f006:**
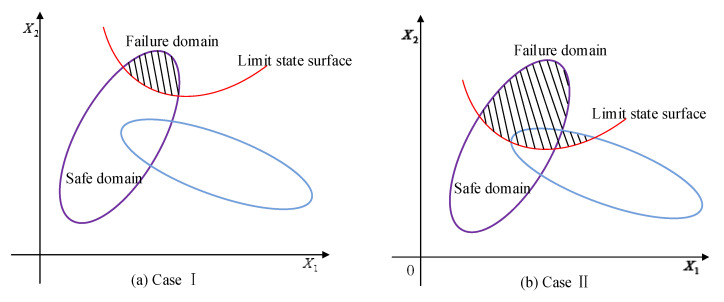
Two different cases of intersection.

**Figure 7 entropy-24-01209-f007:**
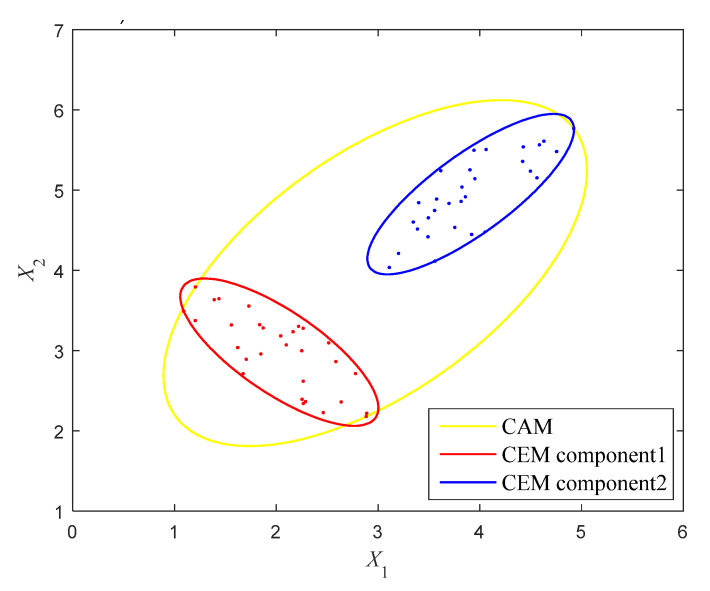
The multi-CEM and CAM of the variables in Example 4.1.

**Figure 8 entropy-24-01209-f008:**
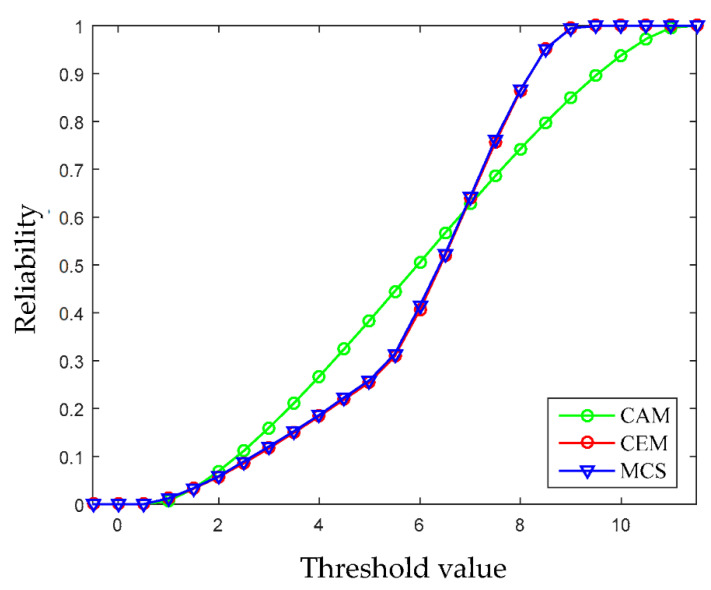
Non-probabilistic reliability under different threshold values (Example 4.1).

**Figure 9 entropy-24-01209-f009:**
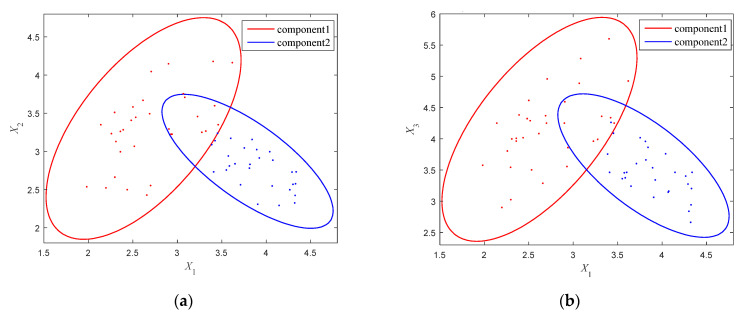
The multi-CEM on the three 2-dimensional planes (Example 4.2). (**a**) *X*_1_-*X*_2_ plane, (**b**) *X*_1_-*X*_3_ plane, (**c**) *X*_2_-*X*_3_ plane.

**Figure 10 entropy-24-01209-f010:**
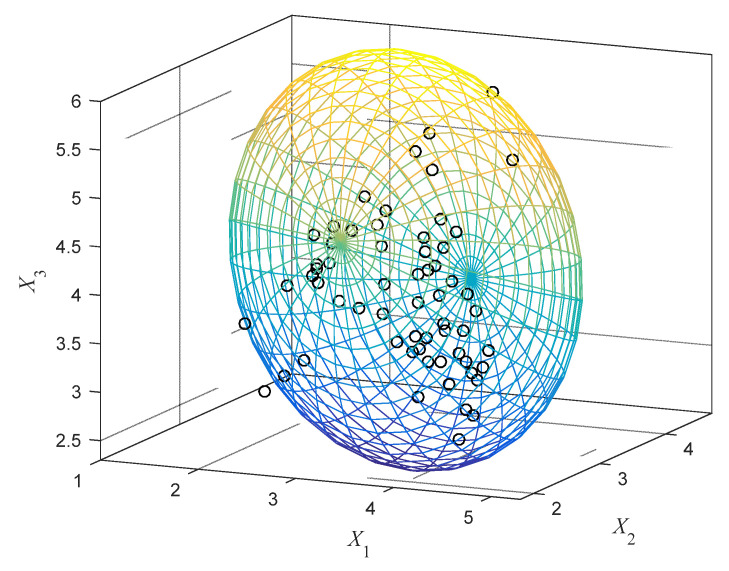
The correlation approximate model of the variables (Example 4.2).

**Figure 11 entropy-24-01209-f011:**
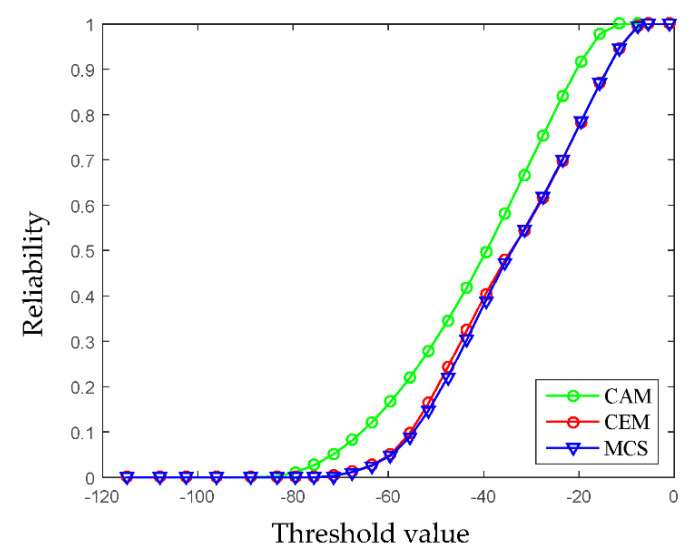
Non-probabilistic reliability under different threshold values (Example 4.2).

**Figure 12 entropy-24-01209-f012:**
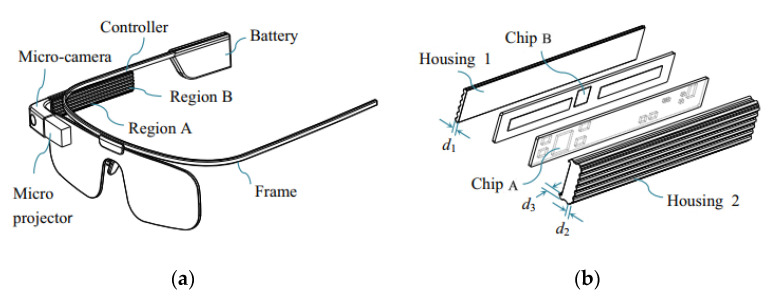
The structure of an augmented reality glasses [34]. (**a**) Components of the augmented reality glasses, (**b**) Exploded diagram of the controller.

**Figure 13 entropy-24-01209-f013:**
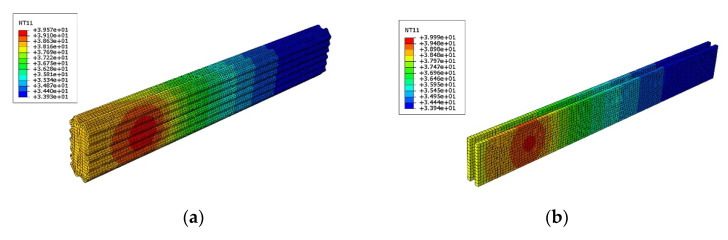
Numerical simulation model of the controller of the AR glasses. (**a**) Surface temperature of the shell, (**b**) Temperature of the circuit board.

**Figure 14 entropy-24-01209-f014:**
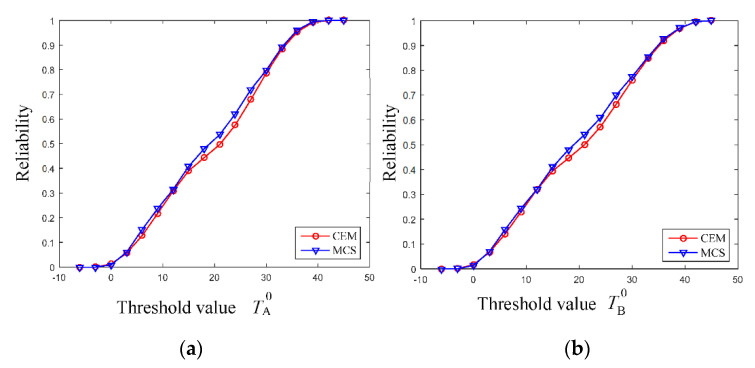
Non-probabilistic reliability under different threshold values. (**a**) Reliability for the temperature at region A, (**b**) Reliability for the temperature at region B.

**Table 1 entropy-24-01209-t001:** Samples of the two uncertain variables (Example 4.1).

No.	*X* _1_	*X* _2_	No.	*X* _1_	*X* _2_
1	2.219399	3.302164	31	4.561986	5.152379
2	2.098374	3.07042	32	3.919348	4.447051
3	1.848857	2.957823	33	4.585504	5.565339
4	2.883305	2.177448	34	3.617155	5.242016
5	2.265046	2.618665	35	4.923524	5.769563
6	2.253232	2.392929	36	3.386686	4.515868
7	2.58532	2.863861	37	4.630296	5.609843
8	2.637891	2.359759	38	3.82427	5.039146
9	1.203835	3.374412	39	3.555722	4.746409
10	2.043340	3.182734	40	4.062128	5.506304
11	1.730330	3.554200	41	4.427443	5.538891
12	2.888488	2.218334	42	3.49437	4.654342
13	2.164607	3.234345	43	3.559939	4.114994
14	1.704952	2.891498	44	3.490927	4.4181
15	2.265823	2.339422	45	3.697514	4.834546
16	1.389795	3.634274	46	4.051388	4.479216
17	2.515282	3.096378	47	3.754507	4.534968
18	1.871032	3.282757	48	3.858271	4.916167
19	1.558403	3.319817	49	3.903247	5.253065
20	1.620956	3.037727	50	4.421196	5.359086
21	2.778897	2.712910	51	3.951404	5.142982
22	2.264103	3.279144	52	3.346107	4.601806
23	1.436786	3.645572	53	3.20174	4.211189
24	1.673714	2.712341	54	3.816983	4.859839
25	1.090868	3.489921	55	4.497481	5.236862
26	1.205242	3.793937	56	3.398991	4.844100
27	2.287038	2.364876	57	3.942939	5.496589
28	2.461811	2.226604	58	3.111521	4.035186
29	1.836134	3.321929	59	3.575249	4.888462
30	2.250109	2.997610	60	4.754638	5.481510

**Table 2 entropy-24-01209-t002:** The optimal parameters of the GMM (Example 4.1).

Parameter	Optimal Value
Weighted averages	$\alpha ={\left[0.4998,0.5002\right]}^{T}$
Covariance matrixes	$\Sigma_{1}=\left[\begin{array}{cc} 0.2444 & -0.1744\\ -0.1744 & 0.2178 \end{array}\right],\Sigma_{2}=\left[\begin{array}{cc} 0.2324 & 0.1843\\ 0.1843 & 0.2254 \end{array}\right]$
Mean values	$\mu_{1}={\left[2.03,2.98\right]}^{T},\;\mu_{2}={\left[3.91,4.95\right]}^{T}$
Critical elliptical contour feature	$R_{1}^{2}=3.884$ $,\;R_{2}^{2}=4.448$

**Table 3 entropy-24-01209-t003:** Samples of the three uncertain variables (Example 4.2).

No.	*X* _1_	*X* _2_	*X* _3_	No.	*X* _1_	*X* _2_	*X* _3_
1	2.5320	3.4462	3.5019	32	3.6189	4.1630	4.9250
2	2.3088	3.1294	3.9969	33	1.9811	2.5370	3.5750
3	3.2268	3.4569	3.9605	34	4.2653	2.4967	3.4031
4	2.3596	3.2615	4.0070	35	3.5516	2.7558	3.3604
5	2.5159	3.0672	4.2894	36	3.8177	2.8293	3.6608
6	2.9055	3.2930	4.5919	37	4.0395	2.9958	3.7589
7	2.2948	3.5102	3.5427	38	4.0746	2.8847	3.1606
8	2.3597	2.9946	3.9629	39	3.5858	2.8088	3.3748
9	2.7068	4.0459	4.9582	40	4.0676	2.5463	3.1449
10	2.9293	3.2253	3.5551	41	3.4528	3.2367	4.0881
11	2.4361	2.4980	4.0164	42	3.6492	2.8374	3.2405
12	2.2589	3.2330	3.8037	43	4.3014	2.5685	2.8392
13	2.3912	3.2835	4.3816	44	3.8116	2.7805	3.9572
14	2.5001	3.5827	4.6131	45	3.7107	2.5622	3.6018
15	2.6889	3.4933	4.3689	46	4.2908	2.7291	3.2746
16	3.0838	3.7081	5.2839	47	3.5738	2.9391	3.4551
17	3.3203	3.2658	4.3593	48	3.3867	3.0881	3.7546
18	2.2970	2.6622	3.0253	49	3.8931	3.0230	3.5358
19	3.0681	3.7557	4.8871	50	4.3254	2.4228	2.9424
20	2.6141	3.6685	4.0826	51	3.7535	3.0447	4.0164
21	2.9403	3.2250	3.8578	52	3.9226	2.9139	3.3404
22	2.6573	2.4284	3.2880	53	4.3303	2.5758	3.2029
23	2.4819	3.4070	4.3218	54	4.3371	2.7318	3.4623
24	3.4196	3.5990	4.3369	55	3.4090	2.7318	3.4623
25	3.2754	3.2505	3.9900	56	3.6011	3.1717	3.4623
26	3.4605	3.3500	4.2500	57	4.1449	2.2918	3.4623
27	2.1395	3.3500	4.2500	58	3.4250	3.1398	4.2623
28	2.9006	4.1481	4.2500	59	4.3210	2.3238	2.6623
29	2.6994	2.5519	4.2500	60	3.8412	3.1557	3.8623
30	3.4030	4.1780	5.6000	61	3.9048	2.3078	3.0623
31	2.1970	2.5220	2.9000				

**Table 4 entropy-24-01209-t004:** The optimal parameters of the GMM (Example 4.2).

Parameter	Optimal Value
Weighted averages	$\alpha ={\left[0.4579,0.5421\right]}^{T}$
Covariance matrixes	$\Sigma_{1}=\left[\begin{array}{c} 0.1477\;\;0.1204\;\;0.1550\\ 0.1204\;\;0.2588\;\;0.2283\\ 0.1550\;\;0.2283\;\;0.3942 \end{array}\right],\Sigma_{2}=\left[\begin{array}{c} 0.1339\;\;-0.0906\;\;-0.1130\\ -0.0906\;\;\;\;0.1113\;\;\;\;0.1181\\ -0.1130\;\;\;\;0.1181\;\;\;\;0.1904 \end{array}\right]$
Mean values	$\mu_{1}={\left[2.62,3.30,4.15\right]}^{T},\;\mu_{2}={\left[3.79,2.87,3.57\right]}^{T}$
Critical elliptical contour feature	$R_{1}^{2}=8.1549$ $,\;R_{2}^{2}=6.9380$

**Table 5 entropy-24-01209-t005:** The optimal parameters for the AEM.

Parameter	Optimal Value
Weighted average	α=1
Covariance matrixes	$\Sigma =\left[\begin{array}{c} 0.0167\;\;-0.0013\;\;-0.0009\\ -0.0013\;\;0.0274\;\;\;\;\;\;0.0180\\ -0.0009\;\;0.0180\;\;\;\;\;\;0.0327 \end{array}\right]$
Mean value	$\mu ={\left[3.0771,\;3.3925,\;4.2394\right]}^{T}$
Compact ratio	$R^{2}=10.082$

## Data Availability

Detailed data are contained within the article.

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
