# Peer review of "Structural Reliability Analysis by Using Non-Probabilistic Multi-Cluster Ellipsoidal Model"

_entropy, 2022, doi:10.3390/e24091209_

Round 1

Reviewer 1 Report

Review comments attached

Reviewer 2 Report

ID: entropy-1828007

Title: Structural reliability analysis by using non-probabilistic multi-cluster ellipsoidal model

This paper proposed an improved method enabling to perform the critical structural reliability analysis having uncontrollable uncertainties by using the non-probabilistic multi-cluster ellipsoidal model. Several model and algorithms are used to construct the multi-CEM.

The validation has been conducted by comparing the proposed method and MCS.

There are some issues to overcome for publication:

  1. There are several equations. However, there some typo exists.

  2. Probabilistic parameters are needed to conduct MCS. Please provide the relevant information.

  3.  

Major :

Line 119,129,131 : It seems that something is missed. Please check it.

Line 142,147 : It seems that something is missed. Please check it.

Line 247,249 : It seems that something is missed. Please check it.

Line 255 : The last term in equation 16 seems to be typo. Please check it.

Line 262 : It seems that something is missed. Please check it.

Line 265 : Please, put a caption for (a) and (b) in Figure 5.

Line 301 : what is the boundary of the variables x1 and x2 ? Also, what is “a” in equation 22 ? Please provide more information.

Line 310 : what is the “COMPACT RATIO” ? and how to compute it ?

Line 330 : Please provide additional information about the probabilistic parameters such as the distribution type, mean, etc of x1 and x2. These information seem to be mandatory information when performing "MCS".

Line 337 & 381 : please provide additional information about the probabilistic parameters of the design variables.

Minor:

Line 193 : what is the “MPP” ?, Is it a most probable point ?

Line 210 : Instead of “Fig.4”, “Figure 4” seems to be proper.

Line 220& 221 : “Beta function” or “beta function”. Please check it.

Line 251 : “Aand” ? it seems to be typo. Please check it.

Reviewer 3 Report

This is an interesting paper but the suggestion, in the introduction, that there are insurmountable problems with conventional probabilistic reliability analysis, is questionable, although analysts will often use some of the methods in this paper in conjunction with conventional analysis.

Because the non-probabilistic convex set method is not well known it would be most helpful if the authors provided a very simple linear example showing the application of the non- probabilistic and probabilistic methods based on the same 'available data. With this, which could be a separate but linked paper,  the paper could be accepted.

Round 2

Reviewer 1 Report

The reviewer's comments and suggestions are incorporated into the revised manuscript. The reviewer does not have further questions and concerns. The reviewer recommends this manuscript be accepted for publication